

# Advancing thyroid diagnosis: integrating AI-driven CAD framework with numerical data and ultrasound images

Saleh Ateeq Almutairi

Department of Computer Science and Information, Applied College, Taibah University, Madinah, Saudi Arabia

## ABSTRACT

This study proposes an advanced computer-aided diagnosis (CAD) framework for thyroid disease diagnosis that integrates numerical patient data and ultrasound images. The framework uses cutting edge technologies, including Vision Transformers (ViTs) and SHapley Additive exPlanations (SHAPs), to increase diagnostic accuracy, interpretability, and clinical applicability. The proposed CAD framework employs the sparse search algorithm (SSA) for optimized feature selection from numerical data and the tree-structured Parzen estimator for tuning the hyperparameters. ViTs are utilized for analyzing thyroid ultrasound images, whereas SHAP provides explainable AI insights into model predictions. Extensive experiments were conducted on two datasets: the thyroid disease patient dataset and the DDTI: Thyroid Ultrasound Images dataset. Performance was evaluated *via* five-fold and ten-fold cross-validation utilizing metrics including accuracy, precision, and recall. The framework achieved promising performance, with models trained without data augmentation consistently outperforming their augmented counterparts. For the thyroid disease patient dataset, the best-performing model reported an accuracy of 99.71%, precision of 97.05%, recall of 99.29%, and F1-score of 98.16%. For the DDTI dataset, ViTs achieved an accuracy of 95.06% without augmentation, surpassing existing methodologies. Key features such as thyroxine, thyroid surgery, and thyroid-stimulating hormone (TSH) were identified as critical predictors of thyroid conditions. This study underscores the potentiality of AI-driven approaches in healthcare, paving the way for improved diagnostic outcomes and personalized treatment strategies.

## INTRODUCTION

Thyroid disorders represent a widespread global health concern, impacting millions of people regardless of age or gender (*Farling, 2000*). Situated in the front of the neck, the thyroid gland is a small, butterfly-shaped organ that produces hormones like thyroxine (T4) and triiodothyronine (T3), which are important for regulating metabolism, growth and energy levels (*Vanderpump, 2011*). When the thyroid gland malfunctions, it can result in various conditions, such as hypothyroidism, hyperthyroidism, thyroid nodules, and

Corresponding author
Saleh Ateeq Almutairi, smoutiri@taibahu.edu.sa

thyroid cancer (*Ahmadi et al., 2020*). Early and precise diagnosis of these disorders is crucial for effective treatment and to prevent potential health complications (*Kratzsch & Pulzer, 2008*). Since the thyroid gland plays a key role in maintaining bodily balance, any impairment in its function can significantly affect an individual's overall health and well-being.

The global burden of thyroid disorders is substantial, with significant implications for public health systems worldwide (*Deng et al., 2020*). According to the American Thyroid Association, over 20M people in the United States (US) alone are affected by thyroid disorders, with up to 60% of cases remaining undiagnosed (*Gessl, Lemmens-Gruber & Kautzky-Willer, 2012*). This high rate of undiagnosed cases underscores the need for improved diagnostic tools and strategies. Women are particularly susceptible, being five to eight times more likely to develop thyroid disorders than men are, a disparity that highlights the importance of gender-specific approaches in thyroid health management (*Castello & Caputo, 2019*). Additionally, the incidence of thyroid cancer has consistently increased in recent years, making it one of the most rapidly growing cancers in terms of new diagnoses (*Antonelli et al., 2015*). This trend necessitates advancements in early detection and treatment methodologies to address the growing prevalence of thyroid cancer.

The diagnosis of thyroid disorders typically involves a multifaceted approach, combining clinical evaluation, thyroid function tests, imaging studies, and, when necessary, biopsy procedures (*Nachiappan et al., 2014*). Thyroid function tests, which measure the levels of thyroid-stimulating hormone (TSH), free thyroxine (FT4), and triiodothyronine (T3), are commonly used to assess thyroid function (*Shukla et al., 2009*). These tests provide critical insights into the hormonal output of the gland and help identify abnormalities in thyroid activity. Imaging techniques, including computed tomography (CT), ultrasound, and magnetic resonance imaging (MRI), are utilized to assess the structure of the thyroid gland and detect abnormalities such as nodules or tumors (*Frunzac & Richards, 2016*). These imaging modalities offer a noninvasive means of visualizing the gland and identifying potential pathologies. Fine-needle aspiration biopsy (FNAB) is often utilized to obtain tissue samples from thyroid nodules for histological analysis, aiding in the exclusion of malignancy (*Wellby, 1976*; *Margret, Lakshmipathi & Kumar, 2012*). This biopsy technique is particularly valuable in distinguishing between benign and malignant nodules, guiding subsequent treatment decisions (*Strauss et al., 2010*).

Treatment strategies for thyroid disorders vary depending on the specific condition and may include medication, radioactive iodine therapy, thyroid surgery, or a combination of these approaches (*Rivkees et al., 2011*). Hypothyroidism is typically controlled by thyroid hormone replacement therapy to restore hormone levels to normal (*Pund et al., 2022*). This treatment approach aims to alleviate symptoms and prevent long-term complications associated with hormone deficiency. Hyperthyroidism may be treated with antithyroid medications, radioactive iodine therapy, or thyroidectomy, which involves partial (or even complete) removal of the thyroid gland (*De Leo, Lee & Braverman, 2016*). Each treatment option has its own set of benefits and risks, requiring careful consideration on the basis of the patient's clinical profile. Thyroid cancer treatment often involves surgical intervention

followed by radioactive iodine therapy and, in some cases, targeted therapy or chemotherapy (*Papaleontiou & Haymart, 2012*). The choice of treatment is influenced by factors such as the cancer stage, patient age, and overall health status, emphasizing the need for personalized treatment plans.

In recent times, artificial intelligence (AI) has become a transformative tool in improving the diagnosis and management of thyroid disorders (*Nagendra, Pappachan & Fernandez, 2023*; *AbdulAzeem et al., 2025b*). AI-based algorithms are capable of processing medical imaging data (*Abd El-Khalek et al., 2024*), including ultrasound or CT scans, to aid in the identification and classification of thyroid nodules and tumors (*Sorrenti et al., 2022*). These systems utilize sophisticated machine learning (ML) methods to detect patterns and irregularities that may signal thyroid-related conditions. Predictive models, developed using ML, can assess demographic, clinical, and laboratory data to identify individuals at higher risk of thyroid disease (*Bini et al., 2021*). Such models facilitate a proactive strategy for managing thyroid health, allowing for early detection and potentially reducing the prevalence of undiagnosed cases. Additionally, AI-powered decision support mechanisms can assist the healthcare staff in analyzing thyroid function tests and imaging results, leading to more precise diagnoses and tailored treatment strategies (*Peng et al., 2021*; *Ludwig et al., 2023*). The integration of AI into healthcare practices enhances diagnostic precision, optimizes workflows, and ultimately improves patient care outcomes.

The major objective of this study is to propose a computer-aided diagnosis (CAD) framework for thyroid diagnosis that integrates both numerical data and ultrasound images from patients with thyroid disease. This framework uses advanced technologies, including vision transformers (ViTs) and SHapley Additive exPlanations (SHAP) explainable AI, to increase the accuracy and interpretability of diagnostic outcomes. ViTs are employed to analyze ultrasound images effectively, aiding in the detection of abnormalities and potential diseases. These transformers utilize self-attention mechanisms to capture global dependencies within the images, enabling comprehensive feature extraction and representation. Additionally, SHAP explainable AI techniques provide transparency to the diagnostic process, enabling clinicians to understand the rationale behind the system's recommendations.

By quantifying the contribution of each input feature to the model's predictions, SHAP enhances the interpretability of AI-driven diagnostic tools, fostering trust and confidence among healthcare providers. The CAD framework also incorporates numerical patient data optimized *via* the Sparrow Search Algorithm to select the most informative features. This optimization process ensures that the model focuses on the most relevant attributes, improving diagnostic performance and reducing computational complexity. Furthermore, this study integrates the tree-structured parameter (TPE) to fine-tune hyperparameters, thereby enhancing overall performance while minimizing computational resources and time. The TPE employs Bayesian optimization to explore the hyperparameter space, identifying configurations that maximize model performance metrics.

The structure of this manuscript is organized as follows: "Related Studies" offers a comprehensive review of pertinent studies, emphasizing significant advancements and methodologies in thyroid disease diagnosis. "Methodology" goes into the proposed

approach, providing a detailed analysis and visual depiction of the methodology, which integrates numerical data and ultrasound images into a CAD framework. "Experiments and Discussion" explores the experiments performed using diverse datasets, accompanied by an analysis of the results and their broader implications. Lastly, "Conclusions and Future Directions" summarizes the findings, discusses their significance, and suggests potential avenues for future research, underscoring the importance of ongoing innovation and clinical validation in the realm of AI-powered thyroid diagnosis.

## RELATED STUDIES

The diagnosis of thyroid disorders has undergone significant advancements in recent years, with numerous studies utilizing various methodologies and technologies to improve diagnostic accuracy and efficiency. Researchers such as *Liu et al. (2023)* and *Bal-Ghaoui et al. (2023)* have made notable contributions to this field, employing innovative approaches to address the challenges associated with thyroid disease diagnosis. Despite these advancements, there remains a research gap in the integration of multiple data modalities, such as numerical patient data and ultrasound images, within a unified framework to increase diagnostic precision and interpretability. This gap raises the need for a more holistic approach that combines the strengths of both data types to improve diagnostic outcomes.

*Margret, Lakshmipathi & Kumar (2012)* explored the use of decision tree attribute splitting criteria for diagnosing thyroid disorders. Their study investigated five distinct splitting criteria for constructing decision trees, including the information gain, likelihood ratio $\chi^2$ statistics, and distance measure. Among these, three criteria are based on impurity measures, whereas the remaining two utilize normalized impurity-based approaches. The decision tree model effectively classified the thyroid dataset into three categories of thyroid illnesses, demonstrating the potential of decision trees in thyroid disease diagnosis. However, these studies focused primarily on numerical data, leaving room for further exploration into the integration of imaging data for a more comprehensive diagnostic approach. This limitation underscores the importance of combining multiple data sources to achieve a more robust diagnostic framework.

In their research, *Margret, Lakshmipathi & Kumar (2012)* and *Shukla et al. (2009)* tackled a significant challenge in medical science by investigating the use of artificial neural networks (ANNs) for diagnosing thyroid disorders. Their study utilized a feed-forward neural network trained with three distinct ANN algorithms, including the backpropagation algorithm (BPA). By evaluating the performance of these algorithms, the research sought to determine the most effective model for thyroid disorder diagnosis. Although the findings demonstrated the promise of ANNs in this domain, the study did not incorporate imaging data, which could enhance diagnostic precision. This gap underscores the importance of adopting a more comprehensive approach that combines both numerical and imaging data for improved accuracy.

Similarly, *Nguyen et al. (2020)* concentrated on creating CAD systems to help clinicians detect thyroid nodules. Their research introduced an AI-based technique for diagnosing malignant thyroid nodules by analyzing data in both spatial and frequency domains. To

tackle the challenge of imbalanced training datasets, the authors employed a weighted binary cross-entropy loss function to train deep convolutional neural networks. Their approach outperformed existing methods when evaluated on the Thyroid Digital Image Database (TDID). However, the study primarily relied on ultrasound images, neglecting the integration of numerical patient data, which could have provided a more comprehensive diagnostic framework. This limitation emphasizes the need to combine imaging data with clinical and laboratory information to enhance diagnostic precision and overall effectiveness.

Building on the application of deep learning (DL), *Liu et al. (2023)* employed advanced techniques to enhance the precision of thyroid cancer diagnosis. The researchers introduced a dynamic integration model named DiTNet, which merges transformer and convolutional neural network (CNN) architectures. This model was specifically designed to distinguish between benign and malignant thyroid nodules. The study utilized data from 202 patients at Quzhou People's Hospital and 102 patients from the publicly available Thyroid Ultrasound Images (DDTI) dataset. When evaluated using receiver operating characteristic (ROC) analysis, DiTNet achieved an area under the curve (AUC) of 95%, along with an accuracy, sensitivity, and specificity of 89%. Although the study highlighted the potential of DL in thyroid cancer diagnosis, it did not incorporate numerical patient data, which could offer valuable diagnostic insights. This limitation indicates that integrating imaging data with clinical information could further improve the assessment performance of AI-based systems.

Similarly, *Bal-Ghaoui et al. (2023)* investigated the diagnostic efficacy of ultrasonography for thyroid and breast cancers, which predominantly affect women worldwide. This study proposed a CNN model for breast ultrasonography and applied it to categorize thyroid nodules. By utilizing shared ultrasound features between the two diseases, researchers have evaluated five pretrained models on both datasets. Their CNN achieved accuracy rates of 0.9484 and 0.8535 for the breast and thyroid datasets, respectively, while maintaining low false-positive rates. Despite these promising results, the study did not incorporate numerical patient data, which could further enhance the diagnostic framework. This omission underscores the need for a more comprehensive approach that integrates both imaging and clinical data.

## Research gap

While the aforementioned studies have made significant strides in thyroid disorder diagnosis, there is a notable gap in the integration of multiple data modalities, such as numerical patient data and ultrasound images, within a unified diagnostic framework. Existing studies often focus on either numerical data or imaging data, but not both. This limitation hinders the development of comprehensive diagnostic tools that can utilize the strengths of both data types. Additionally, there is a need for more interpretable AI models that can provide clinicians with transparent insights into the diagnostic process.

The current study aims to address these gaps by proposing a CAD framework that integrates numerical data and ultrasound images, utilizing advanced technologies such as ViTs and SHAP explainable AI to increase diagnostic accuracy and interpretability. By

bridging these gaps, this study seeks to advance the field of thyroid diagnosis and provide clinicians with a more robust and transparent diagnostic tool.

## METHODOLOGY

This study proposes a CAD framework (see Fig. 1) for thyroid disease diagnosis, which uses both numerical patient data and ultrasound images. The framework integrates ViTs and SHAP explainable AI to increase diagnostic accuracy and interpretability. ViTs, renowned for their exceptional performance in image classification tasks, are employed to analyze thyroid ultrasound images, facilitating the identification of abnormalities and potential diseases.

The incorporation of SHAP explainable AI techniques introduces a layer of transparency into the diagnostic process. SHAP enables the interpretation of model predictions by attributing them to the contribution of each input feature, thereby providing valuable insights into the decision-making process of the CAD system. This transparency not only enhances the trustworthiness of diagnostic results but also allows clinicians to understand the rationale behind the system's recommendations.

In addition to ultrasound images, the CAD framework incorporates numerical data from thyroid disease patients. This study employs the sparse search algorithm (SSA) to optimize the selection of the most promising features from these data. The SSA efficiently searches the feature space to determine the most informative characteristics for precise diagnosis. By utilizing SSA, the CAD system can effectively handle high-dimensional numerical data and extract meaningful insights, thereby improving diagnostic performance.

Furthermore, the TPE is integrated to fine-tune the hyperparameters of the CAD framework and enhance its overall performance. It is a Bayesian-based optimization technique that systematically searches the hyperparameter space to identify configurations that maximize the model's performance metrics. By utilizing it, the CAD system can adaptively adjust its parameters to achieve optimal diagnostic accuracy while minimizing computational resources and time.

## MATERIALS

The study uses two primary datasets: the thyroid disease patient dataset and the DDTI: Thyroid Ultrasound Images dataset. The thyroid disease patient dataset encompasses a wide range of attributes, including demographic information (*e.g.*, age, sex), medical history (*e.g.*, thyroxine intake, antithyroid medications, past surgeries), and current health status (*e.g.*, presence of goiter, tumor, or hypopituitary conditions). Laboratory results such as TSH, T3, TT4, T4U, and FTI levels are also included. This dataset is accessible at https://www.kaggle.com/datasets/kapoorprakhar/thyroid-disease-patient-dataset.

The DDTI dataset, an open-access resource supported by Universidad Nacional de Colombia, CIM@LAB, and IDIME, comprises 99 cases and 134 ultrasound images. Each case includes an XML file with expert annotations and patient information. The dataset is available at https://www.kaggle.com/datasets/dasmehdixtr/ddti-thyroid-ultrasound-images.
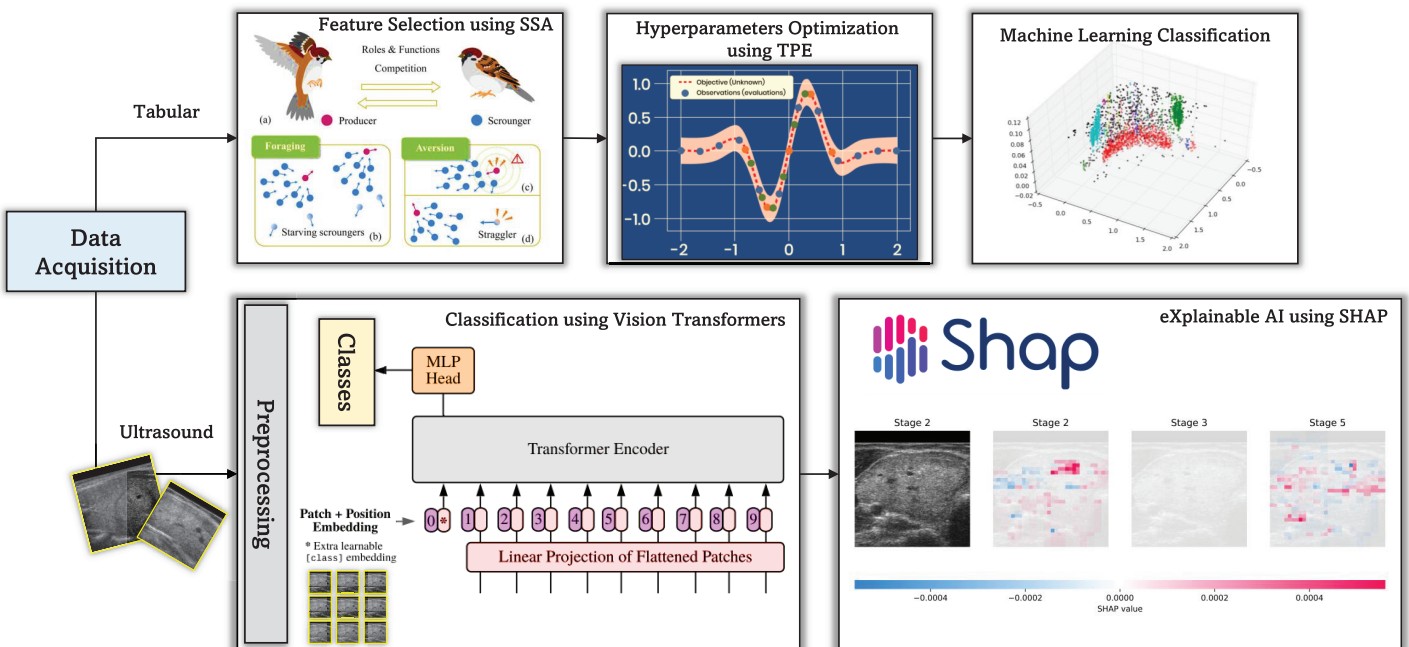

**Figure 1** The proposed CAD framework for thyroid diagnosis utilizing tabular and image data.

## Numerical feature selection *via* the sparrow search algorithm

Feature selection is a useful step in building effective ML models, particularly for medical diagnosis tasks such as thyroid disease classification (*Balaha, Hassan & Balaha, 2025*). The SSA, inspired by the foraging behavior of sparrows, is employed to optimize feature selection. The SSA mimics the exploration and exploitation strategies of sparrows to search efficiently for the most informative features (*Xue & Shen, 2020*).

The primary objective of feature selection *via* SSA is to identify the subset of input features that have the greatest impact on the prediction performance of the ML model. This approach reduces the dimensionality of the feature space, improving model interpretability and reducing computational complexity (*Gad et al., 2022*).

Mathematically, the SSA operates by iteratively updating the position of a population of candidate solutions (sparrows) in the search space. The position of each sparrow represents a potential solution, *i.e.*, a subset of features selected for classification. The movement of sparrows is guided by a combination of exploration and exploitation strategies, aiming to balance the search for new regions and the exploitation of promising solutions (*Gharehchopogh et al., 2023*).

The movement of sparrows in SSA is governed by Eqs. (1) and (2), where $Pos(t)$ represents the position of sparrows at iteration $t$, $StepSize(t)$ denotes the step size controlling the magnitude of movement, Rand is a random number sampled from a uniform distribution, $\lambda$ is a scaling factor, $\beta$ is a control parameter, and $\Gamma$ denotes the gamma function. Algorithm 1 summarizes the major steps for feature selection *via* the SSA.

---

**Algorithm 1 Pseudocode of SSA for feature selection.**

1 Initialize the population of sparrows with random feature subsets.

2 Initialize parameters: StepSize$_{max}$, StepSize$_{min}$, λ, β, MaxIter, NumFeatures.

3 **for** *each iteration t until convergence* **do**

4       Update step size: $\text{StepSize}(t) = \text{StepSize}_{min} + (\text{StepSize}_{max} - \text{StepSize}_{min}) \times \frac{t}{\text{MaxIter}}$.

5       Generate Levy flight: $\text{LevyFlight} = \text{Rand} \times \times \left( \frac{\lambda \times \Gamma(1+\beta) \times \sin(\pi \times \beta/2)}{\Gamma((1+\beta)/2) \times \beta \times 2^{(\beta-1)/2}} \right)^{1/\beta}$.

6       Update sparrow positions: $\text{Pos}(t+1) = \text{Pos}(t) + \text{StepSize}(t) \times \text{LevyFlight}$.

7       Evaluate the fitness of each sparrow *via* a feature selection metric (*e.g.*, classification accuracy).

8       The global best solution is updated if the current solution is better.

9       Update the local best solution for each sparrow.

10 **end**

---

$$\text{Pos}(t+1) = \text{Pos}(t) + \text{StepSize}(t) \times \text{LevyFlight} \tag{1}$$

$$\text{LevyFlight} = \text{Rand} \times \left( \frac{\lambda \times \Gamma(1+\beta) \times \sin(\pi \times \beta/2)}{\Gamma((1+\beta)/2) \times \beta \times 2^{(\beta-1)/2}} \right)^{1/\beta}. \tag{2}$$

## Numerical classification *via* machine learning

Machine learning techniques play a pivotal role in thyroid disease classification by utilizing patient data to distinguish between benign *vs.* malignant cases. A diverse range of classifiers, including support vector machine (SVM), random forest (RF), AdaBoost, LightGBM (LGBM), and logistic regression (LR), are employed to achieve accurate and reliable predictions (*Song, Ristenpart & Shmatikov, 2017*; *Balaha et al., 2024b*). To increase the performance of these classifiers, various scaling techniques are applied to preprocess the input features. These include robust scaling, min–max scaling (MinMax), max absolute scaling (MaxAbs), standardization (STD), L1 normalization (L1), L2 normalization (L2), and max normalization (Max). These scaling techniques ensure that the input features are appropriately scaled and centered, thereby improving the convergence and stability of the ML algorithms (*Ahsan et al., 2021*).

Additionally, the TPE is employed to optimize the hyperparameters of the classifiers and scalers. It is a Bayesian-based optimization technique that efficiently searches the hyperparameter space to identify configurations that maximize the model's performance metrics (*Watanabe, 2023*). The optimization process involves maximizing the expected improvement over the current best solution, as formulated in Eq. (3), where $x$ represents the hyperparameter configuration, $\mu(x)$ and $\sigma(x)$ are the mean and standard deviation of the predictive distribution, respectively, $f(x^*)$ represents the best-observed value thus far, and $\zeta$ is a tunable exploration-exploitation trade-off parameter.

$$EI(x) = \begin{cases} \frac{\mu(x) - f(x^*) - \zeta}{\sigma(x)}, & \text{if } \sigma(x) > 0 \\ 0, & \text{otherwise} \end{cases}. \tag{3}$$

By iteratively evaluating the performance of different hyperparameter configurations and updating the surrogate model, TPE guides the search toward regions of the hyperparameter space that are likely to yield better results, ultimately improving the classification accuracy of the models.

## Imaginary classification *via* vision transformers

ViTs are promising approaches for classification tasks in thyroid diagnosis *via* ultrasound images. Initially designed for natural image classification, ViTs have demonstrated remarkable adaptability to medical image analysis domains (*Park & Kim, 2022*). The core of ViTs lies in their ability to process input images through a series of self-attention mechanisms, enabling comprehensive feature extraction and representation (*Raghu et al., 2021*). Mathematically, the mechanism of the self-attention ($\mathcal{A}(Q, K, V)$) can be expressed as shown in Eq. (4), where $Q$, $K$, and $V$ correspond to the query, key, and value matrices, respectively; $d_k$ represents the dimensionality of the key vectors; and the SoftMax function calculates the weights of the attention vectors. These weights are subsequently multiplied by the value vectors to produce the attention output (*Zhou et al., 2022*).

$$\mathcal{A}(Q, K, V) = \text{SoftMax}\left(\frac{Q \times K^T}{\sqrt{d_k}}\right) \times V. \tag{4}$$

In the context of thyroid diagnosis, ViTs can effectively distinguish between benign and malignant thyroid nodules by processing ultrasound images through multiple layers of self-attention blocks. These blocks capture both low-level image features and high-level contextual information, enabling the model to learn hierarchical representations of the data. Each self-attention block comprises multiple attention heads that support the model to simultaneously attend to different regions of the image (*Khan et al., 2022*).

The feature representation obtained from the self-attention blocks is further refined through feed-forward neural network layers, facilitating classification. Mathematically, the feed-forward transformation is represented in Eq. (5), where $x$ represents the input feature representation; $W_1$, $b_1$, $W_2$, and $b_2$ denote the weights and biases of the feed-forward neural network layers, and ReLU denotes the rectified linear unit activation function (*Jamil, Jalil Piran & Kwon, 2023*).

$$\text{FFN}(x) = \text{ReLU}(x \times W_1 + b_1) \times W_2 + b_2. \tag{5}$$

Through extensive training on large-scale datasets, ViTs can learn discriminative features indicative of pathological characteristics associated with thyroid malignancy, such as irregular borders, microcalcifications, and increased vascularity.

## eXplainable AI using SHAP: enhancing medical understanding of thyroid conditions

Explainable artificial intelligence (XAI) employing SHAP provides invaluable insights into the decision-making processes of the ML models, especially in healthcare (*Arrieta et al., 2020*). SHAP values offer a comprehensive view of how individual features contribute to model predictions, enhancing interpretability and trust (*García & Aznarte, 2020*; *Aljadani et al., 2023*).

Mathematically, SHAP values are represented as in Eq. (6), where $\phi_i(f)$ represents the SHAP value of feature $i$ for prediction $f$, $F$ denotes the set of all features, $S$ represents a subset of features excluding feature $i$, and $f(S)$ and $f(S \cup \{i\})$ denote the model's output for subsets $S$ and $S$ with feature $i$ included, respectively (*Speith, 2022*).

$$\phi_i(f) = \sum_{S \subseteq F \setminus \{i\}} \frac{|S|!(|F| - |S| - 1)!}{|F|!} \times [f(S \cup \{i\}) - f(S)]. \tag{6}$$

From a medical perspective, SHAP analysis reveals the significance of various factors in diagnosing and managing thyroid conditions. For example, in assessing thyroid disorders such as hypothyroidism or hyperthyroidism, SHAP highlights the contributions of crucial biomarkers such as TSH and thyroxine levels and patient demographics. By quantifying the impact of each feature on model predictions, SHAP enables clinicians to prioritize relevant clinical markers and tailor treatment strategies accordingly (*Tjoa & Guan, 2020*).

Moreover, SHAP-based explanations facilitate patient–centered care by empowering individuals to understand the rationale behind diagnostic decisions. Patients can learn how their medical history, laboratory results, and demographic characteristics influence the likelihood of having thyroid disease. This transparency fosters trust between patients and healthcare providers, ultimately leading to more collaborative decision-making processes and improved health outcomes.

## Evaluation method

To assess the performance of ML models in classifying thyroid diseases, a range of evaluation metrics are utilized, such as accuracy (ACC), balanced accuracy (BAC), precision (PRC), recall (REC), specificity (SPC), F1-score, and intersection over union (IoU). Each metric offers distinct insights into different facets of the model's classification performance (*AbdulAzeem et al., 2025a*). Accuracy reflects the overall correctness of the model's predictions, while BAC addresses class imbalance by averaging the accuracy of both classes. Precision evaluates the ratio of true positive predictions among all positive predictions, and recall (or sensitivity) measures the ratio of true positives identified out of all actual positive cases (*Balaha et al., 2025*). Specificity assesses the model's ability to classify negative cases correctly, and the F1-score provides a harmonic balance between precision and recall. Lastly, IoU measures the overlap between predicted and ground truth labels, making it particularly valuable for tasks like semantic segmentation (*Adhikari et al., 2021*).

To ensure the robustness of the performance evaluation, cross-validation is employed with both 5-fold and 10-fold schemes. Each cross-validation iteration divides the dataset into multiple subsets, with a portion reserved for training and the rest reserved for validation. This process is repeated numerous times, with different subsets used for training and validation in each iteration. The model's generalization ability can be reliably assessed by averaging the performance metrics across multiple cross-validation folds (*Balaha et al., 2024a*).

To account for variability in performance due to randomness in the data and model initialization, 100 trials are conducted for each cross-validation scheme. This extensive

experimentation enables the estimation of 95% confidence intervals (CIs) for each performance metric, providing a measure of the uncertainty associated with the reported results. Through this rigorous evaluation process, the reliability and generalization ability of ML models for thyroid disease classification can be accurately assessed, facilitating informed clinical decision-making and improving patient care (*Kim, Hong & Yoon, 2020*).

### Training details of vision transformers

The ViTs were trained using the following setup:

– Pretraining details: The ViT models were pretrained on large-scale datasets such as ImageNet-21k to utilize transfer learning. This pretraining step helps the model capture generic image features before fine-tuning on the specific task of thyroid ultrasound image classification.

– Fine-tuning parameters: (1) Learning rate: We employed a cosine annealing learning rate scheduler with an initial learning rate of 0.001. The learning rate was gradually reduced over the course of training to ensure convergence. (2) Number of epochs: The models were fine-tuned for a total of 50 epochs. (3) Early stopping with a patience of 10 epochs was implemented to prevent overfitting and to optimize training time. (4) Batch size: A batch size of 32 was used during fine-tuning to balance computational efficiency and model performance.

– Regularization techniques: (1) Weight decay: L2 regularization (weight decay) was applied with a coefficient of 0.0001 to penalize large weights and improve generalization. (2) Dropout: Dropout layers with a dropout rate of 0.1 were incorporated within the transformer blocks to further mitigate overfitting. (3) Data augmentation: Although data augmentation led to decreased performance in our experiments (as discussed in the Experiments section), standard augmentations such as random horizontal flipping and minor rotations were initially considered but later excluded from the final model training due to their negative impact on accuracy.

– Optimizer: Adam optimizer was utilized for its adaptive learning rate properties, which are beneficial for training deep neural networks like ViTs.

– Loss function: Cross-entropy loss was used as the objective function to optimize the classification performance of the ViT models.

– Hardware and software: The models were trained on an NVIDIA GPU with 6 GB of memory, utilizing Python and relevant deep learning libraries such as PyTorch for implementation.

## EXPERIMENTS AND DISCUSSION

### Computing infrastructure

The experiments for this study were designed to rigorously evaluate the performance of the proposed CAD framework for thyroid disease diagnosis. The software configuration was centered on Python, utilizing Windows 11 as the operating system and Anaconda as the distribution platform. The hardware specifications included an NVIDIA GPU with 6 GB of

**Table 1 Performance metrics (average with 95% confidence interval) from feature selection *via* 10-fold cross-validation after 100 trials on the thyroid disease patient dataset.**

| Approach | ACC | BAC | PRC | REC | SPC | F1 | IoU |
|---|---|---|---|---|---|---|---|
| SVM | 98.46% ± 0.0 | 91.29% ± 0.0 | 96.79% ± 0.0 | 82.82% ± 0.0 | 99.77% ± 0.0 | 89.26% ± 0.0 | 80.6% ± 0.0 |
| RF | 99.66% ± 0.0 | 99.46% ± 0.0001 | 96.51% ± 0.0005 | 99.23% ± 0.0002 | 99.7% ± 0.0 | 97.85% ± 0.0003 | 95.79% ± 0.0005 |
| AdaBoost | 99.6% ± 0.0 | 98.68% ± 0.0 | 97.26% ± 0.0 | 97.59% ± 0.0 | 99.77% ± 0.0 | 97.43% ± 0.0 | 94.98% ± 0.0 |
| LGBM | 99.6% ± 0.0 | 99.0% ± 0.0 | 96.62% ± 0.0 | 98.28% ± 0.0 | 99.71% ± 0.0 | 97.44% ± 0.0 | 95.02% ± 0.0 |
| XGB | 99.52% ± 0.0 | 98.17% ± 0.0 | 97.23% ± 0.0 | 96.56% ± 0.0 | 99.77% ± 0.0 | 96.9% ± 0.0 | 93.98% ± 0.0 |
| HGB | 99.55% ± 0.0 | 98.97% ± 0.0 | 95.97% ± 0.0 | 98.28% ± 0.0 | 99.66% ± 0.0 | 97.11% ± 0.0 | 94.39% ± 0.0 |
| MLP | 99.28% ± 0.0001 | 97.56% ± 0.0006 | 95.17% ± 0.0009 | 95.52% ± 0.0012 | 99.59% ± 0.0001 | 95.34% ± 0.0008 | 91.1% ± 0.0014 |
| KNN | 98.12% ± 0.0 | 90.0% ± 0.0 | 94.35% ± 0.0 | 80.41% ± 0.0 | 99.6% ± 0.0 | 86.83% ± 0.0 | 76.72% ± 0.0 |
| DT | 99.67% ± 0.0001 | 99.1% ± 0.0002 | 97.32% ± 0.0006 | 98.42% ± 0.0004 | 99.77% ± 0.0001 | 97.87% ± 0.0003 | 95.82% ± 0.0006 |
| ET | 94.01% ± 0.0003 | 61.48% ± 0.0018 | 97.17% ± 0.002 | 23.02% ± 0.0035 | 99.94% ± 0.0 | 37.18% ± 0.0047 | 22.86% ± 0.0035 |
| LR | 97.32% ± 0.0 | 87.84% ± 0.0 | 87.11% ± 0.0 | 76.63% ± 0.0 | 99.05% ± 0.0 | 81.54% ± 0.0 | 68.83% ± 0.0 |
| Best | 99.71% ± 0.0 | 99.52% ± 0.0001 | 97.05% ± 0.0005 | 99.29% ± 0.0002 | 99.75% ± 0.0 | 98.16% ± 0.0003 | 96.38% ± 0.0005 |

memory, 128 GB of RAM, and an Intel Core i7 processor, ensuring sufficient computational power for training and evaluating DL models.

## Experiments on the numerical thyroid disease patient dataset

Table 1 presents the performance metrics obtained from feature selection *via* a 10-fold cross-validation setup. The metrics are represented as the mean values with 95% confidence intervals (CIs) derived from 100 trials on the thyroid disease patient dataset. The "Best" row highlights the performance metrics from the top-performing models employing majority voting.

Among the evaluated algorithms, AdaBoost, decision tree (DT), histogram-based gradient boosting (HGB), logistic regression (LR), and random forest (RF) emerged as the top performers. These models were optimized with specific hyperparameters, such as the regularization parameter $C = 2.0288193$ for logistic regression and the learning rate $\eta = 0.193226797$ for AdaBoost. The decision tree was configured with entropy as the splitting criterion, a maximum depth of 4, and the best splitter. The random forest algorithm adopted a Gini criterion, a maximum depth of 8, and 99 estimators, whereas the HGB algorithm employed a learning rate of $\eta = 0.3412151542$ and a maximum depth of 10. The number of features selected varied across the models, with logistic regression utilizing nine features, AdaBoost and random forest employing 13 features each, and decision tree and HGB using 16 features. Scaling techniques further enhanced model performance, with logistic regression using L1 normalization, AdaBoost employing MinMax scaling, decision tree utilizing MaxAbs scaling, and both random forest and HGB employing robust scaling.

Table 2 presents similar results for 5-fold cross-validation. The top-performing algorithms include DTs, multilayer perceptrons (MLPs), RFs, and support vector machines (SVMs). These models were assessed on the basis of their classification accuracy and robustness across different hyperparameter configurations. The optimal hyperparameters

**Table 2 Performance metrics (average with 95% confidence interval) from feature selection *via* 5-fold cross-validation after 100 trials on the thyroid disease patient dataset.**

| Approach | ACC | BAC | PRC | REC | SPC | F1 | IoU |
|---|---|---|---|---|---|---|---|
| SVM | 98.49% ± 0.0 | 94.14% ± 0.0 | 91.2% ± 0.0 | 89.0% ± 0.0 | 99.28% ± 0.0 | 90.09% ± 0.0 | 81.96% ± 0.0 |
| RF | 99.63% ± 0.0001 | 99.45% ± 0.0001 | 96.12% ± 0.0008 | 99.23% ± 0.0002 | 99.66% ± 0.0001 | 97.65% ± 0.0004 | 95.41% ± 0.0008 |
| AdaBoost | 99.6% ± 0.0 | 98.68% ± 0.0 | 97.26% ± 0.0 | 97.59% ± 0.0 | 99.77% ± 0.0 | 97.43% ± 0.0 | 94.98% ± 0.0 |
| LGBM | 99.52% ± 0.0 | 98.95% ± 0.0 | 95.65% ± 0.0 | 98.28% ± 0.0 | 99.63% ± 0.0 | 96.95% ± 0.0 | 94.08% ± 0.0 |
| XGB | 99.44% ± 0.0 | 98.28% ± 0.0 | 95.92% ± 0.0 | 96.91% ± 0.0 | 99.66% ± 0.0 | 96.41% ± 0.0 | 93.07% ± 0.0 |
| HGB | 99.52% ± 0.0 | 98.48% ± 0.0 | 96.59% ± 0.0 | 97.25% ± 0.0 | 99.71% ± 0.0 | 96.92% ± 0.0 | 94.02% ± 0.0 |
| MLP | 99.22% ± 0.0001 | 97.04% ± 0.0005 | 95.44% ± 0.0008 | 94.46% ± 0.001 | 99.62% ± 0.0001 | 94.95% ± 0.0007 | 90.38% ± 0.0012 |
| KNN | 98.57% ± 0.0 | 93.08% ± 0.0 | 94.38% ± 0.0 | 86.6% ± 0.0 | 99.57% ± 0.0 | 90.32% ± 0.0 | 82.35% ± 0.0 |
| DT | 99.59% ± 0.0001 | 98.95% ± 0.0002 | 96.63% ± 0.001 | 98.18% ± 0.0005 | 99.71% ± 0.0001 | 97.39% ± 0.0005 | 94.92% ± 0.001 |
| ET | 94.55% ± 0.0003 | 65.04% ± 0.0019 | 97.36% ± 0.0012 | 30.15% ± 0.0037 | 99.93% ± 0.0 | 46.01% ± 0.0043 | 29.9% ± 0.0037 |
| LR | 96.79% ± 0.0 | 80.94% ± 0.0 | 94.27% ± 0.0 | 62.2% ± 0.0 | 99.68% ± 0.0 | 74.95% ± 0.0 | 59.93% ± 0.0 |
| Best | 99.69% ± 0.0001 | 99.23% ± 0.0002 | 97.28% ± 0.0011 | 98.69% ± 0.0004 | 99.77% ± 0.0001 | 97.98% ± 0.0006 | 96.04% ± 0.0011 |

for each model were determined through extensive experimentation, with MLP employing a ReLU activation function, 256 hidden layers, and an Adam solver; RF utilizing a Gini criterion, a maximum depth of 8, and 86 estimators; DT employing a Gini criterion, a maximum depth of 18, and the best splitter; and SVM utilizing a radial basis function kernel with a regularization parameter $C = 1.59$.

Figure 2 visualizes the feature importance of the 5- and 10-fold approaches for the thyroid disease patient dataset. The features were assessed for their importance in predicting thyroid conditions. Among the features, "thyroxine", "thyroidsurgery", and "TSH" were identified as the most important, each with an importance score of $>80\%$ for ten folds and $>65\%$ for five folds. Similarly, "T4" exhibited less importance between 60% and 70% for the ten folds and between 50% and 60% for the five folds.

The results highlight the pivotal role of "thyroxine", "thyroidsurgery", and "TSH" from the medical perspective of thyroid conditions. Thyroxine ($T_4$) is a hormone crucial for regulating metabolism, energy production, and growth. Its high importance in predicting thyroid conditions suggests its importance in diagnosing and managing thyroid disorders, such as hypothyroidism or hyperthyroidism. Thyroidsurgery indicates a history of surgical intervention, possibly due to thyroid nodules, tumors, or other thyroid-related pathologies. This feature underscores the clinical importance of past surgical procedures in understanding and treating thyroid disorders. TSH, a thyroid-stimulating hormone, is a key marker of thyroid function, with elevated levels indicative of hypothyroidism and suppressed levels suggestive of hyperthyroidism. Its prominence underscores its fundamental role in thyroid diagnosis and monitoring.

### Experiments on the DDTI: thyroid ultrasound images dataset

Table 3 lists the performance metrics derived from the DDTI (Thyroid Ultrasound Images) dataset employing ViTs, with a particular focus on the impact of data

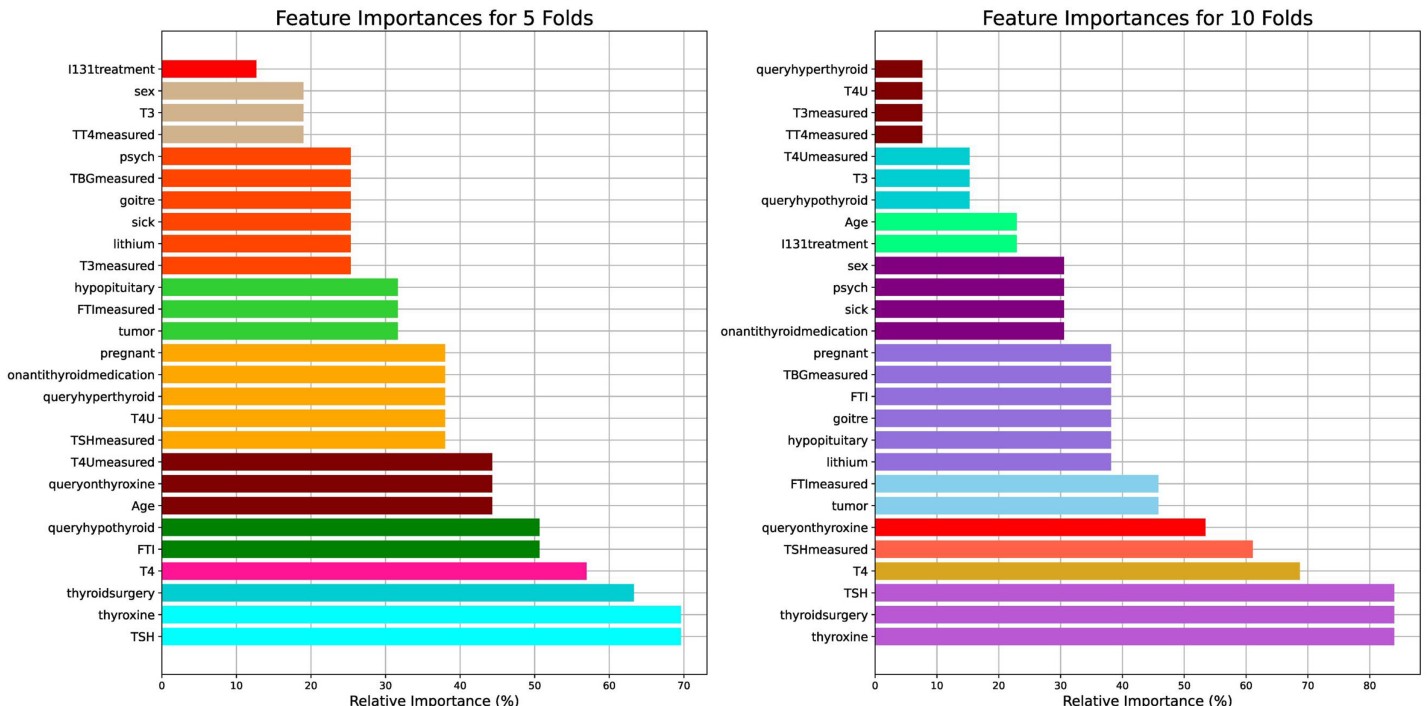

**Figure 2** Visual presentation of the feature importance for the 5- and 10-fold comparisons with respect to the thyroid disease patient dataset.

**Table 3 Performance metrics resulting from DDTI: thyroid ultrasound image dataset using ViTs (using base-32 model).**

| Model | Augmentation | ACC (%) | PRC (%) | REC (%) | SPC (%) | F1 (%) | IoU (%) | BAC (%) | MCC (%) | Youden (%) | Yule (%) |
|---|---|---|---|---|---|---|---|---|---|---|---|
| P16-224-In21K | ✗ | 92.02 | 82.21 | 80.69 | 94.69 | 81.03 | 68.43 | 87.69 | 76.17 | 75.39 | 97.36 |
| P16-224 | ✗ | 95.00 | 88.24 | 87.90 | 96.47 | 87.93 | 78.53 | 92.18 | 84.77 | 84.36 | 99.09 |
| P32-384 | ✗ | 95.06 | 88.19 | 87.61 | 96.53 | 87.65 | 78.11 | 92.07 | 84.61 | 84.13 | 99.14 |
| P16-224-In21K | ✓ | 68.64 | 55.33 | 28.53 | 73.93 | 17.58 | 10.71 | 51.23 | 3.47 | 2.46 | 50.49 |
| P16-224 | ✓ | 74.49 | 43.21 | 38.62 | 81.44 | 37.44 | 23.24 | 60.03 | 22.48 | 20.06 | 54.87 |
| P32-384 | ✓ | 75.71 | 44.35 | 40.63 | 82.45 | 39.34 | 24.77 | 61.54 | 25.11 | 23.08 | 60.15 |

augmentation. The models are categorized on the basis of whether augmentation was employed or not, and various evaluation metrics, including accuracy, precision, recall, specificity, and Yule's Q coefficient, are presented.

In analyzing the outcomes, it becomes evident that models trained without data augmentation consistently outperform their augmented counterparts, particularly in the context of ultrasound imagery. Notably, the models denoted as P16-224 and P32-384 exhibit superior performance, boasting accuracy scores of 95.00% and 95.06%, respectively, alongside commendable precision, recall, specificity, and F1-scores. These results underscore the efficacy of these models in accurately classifying thyroid ultrasound images without the need for data augmentation.

However, upon introducing augmentation (✓), a significant decline in performance is observed across all the metrics. For example, the accuracy decreases to 68.64%, 74.49%, and 75.71% for the augmented models P16-224-In21K, P16-224, and P32-384, respectively. This decline raises pertinent concerns regarding the applicability and efficacy of data augmentation in the domain of ultrasound imaging.

One of the primary concerns revolves around the intricate nature of ultrasound data. Ultrasound images inherently possess unique characteristics, such as varying tissue densities, acoustic impedance, and imaging artifacts, which can hinder traditional augmentation techniques. Moreover, the clinical relevance and diagnostic accuracy of ultrasound images necessitate preserving the integrity and fidelity of the data, making it imperative to tread cautiously when applying augmentation.

Furthermore, the efficacy of augmentation techniques in enhancing model generalizability and robustness may be contingent upon the appropriateness of the augmentation strategies employed. Given the nuanced nature of ultrasound imaging, conventional augmentation techniques may inadvertently introduce distortions or artifacts that compromise the interpretability and diagnostic utility of the images. Consequently, a delicate balance exists between augmenting the data to improve model performance and preserving the inherent characteristics of ultrasound images.

Figure 3 illustrates the interpretability achieved through SHAP on DDTI: the Thyroid Ultrasound Images dataset. The visualization shows two randomly selected samples from the dataset, each accompanied by the top three predictions generated by the model. Additionally, the figure provides insights into the interpretability of these predictions, offering a transparent view of the features contributing most significantly to each prediction. This comprehensive approach enhances the understanding of the model's decision-making process and facilitates the identification of crucial diagnostic factors in thyroid ultrasound images.

The current study outperforms the related studies by Liu et al. (2023) and Bal-Ghaoui et al. (2023) regarding diagnostic accuracy for thyroid disorders via ultrasound imaging. Liu et al. (2023) achieved an AUC of 0.95 in receiver operating characteristic (ROC) analysis with their dynamic integration model. Bal-Ghaoui et al. (2023) attained high accuracy rates of 94.84% and 85.35% for breast and thyroid datasets, respectively, with their customized CNN architecture; the current study achieved even higher accuracy rates. Specifically, models trained without data augmentation consistently demonstrated superior performance across various evaluation metrics, with accuracy scores reaching 95.06%. These results highlight the effectiveness of the suggested approach in accurately classifying thyroid ultrasound images, surpassing the performance of existing methodologies in the field.

## Clinical relevance

The experiments conducted in this study hold significant clinical relevance, particularly in the context of improving diagnostic accuracy and interpretability for thyroid disorders. Thyroid diseases, including hypothyroidism, hyperthyroidism, thyroid nodules, and thyroid cancer, pose substantial challenges to healthcare systems worldwide due to their

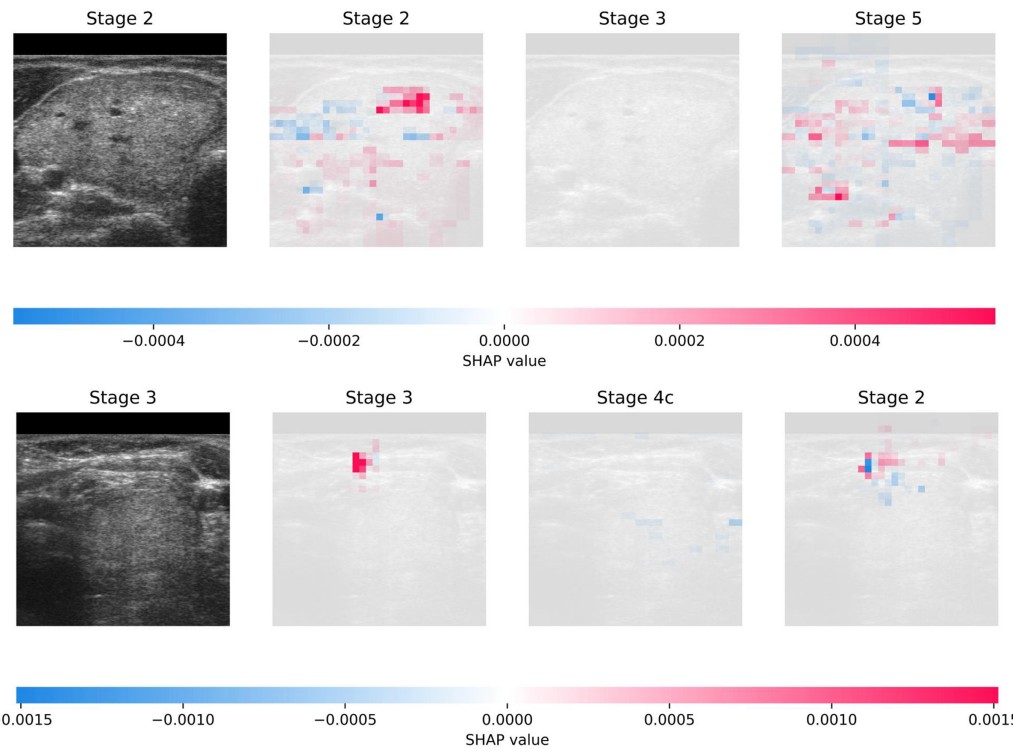

**Figure 3** **Visual presentation of explainability using SHAP on DDTI: Thyroid Ultrasound Image dataset.**

high prevalence and potential impact on patient health. Early and precise diagnosis is critical for effective treatment and the prevention of long-term complications. The integration of numerical patient data and ultrasound images into a unified CAD framework represents a major advancement in addressing these challenges. By utilizing advanced technologies such as ViTs and SHAP, the proposed framework not only enhances diagnostic accuracy but also provides transparency in the decision-making process, fostering trust among clinicians and patients.

From a clinical perspective, the use of ViTs for analyzing thyroid ultrasound images is particularly noteworthy. Ultrasound imaging is a non-invasive and widely used modality for evaluating thyroid conditions, but its interpretation can be subjective and prone to variability among radiologists. The ability of ViTs to capture global dependencies within images and extract hierarchical features enables the model to identify subtle abnormalities that may be missed by human observers. For instance, the detection of irregular borders, microcalcifications, and increased vascularity (key indicators of malignancy) is significantly improved through the use of ViTs. This capability aligns with the clinical need for tools that can assist radiologists in making more accurate and consistent diagnoses, thereby reducing the risk of false positives and false negatives.

The incorporation of SHAP into the framework further enhances its clinical utility by providing interpretable insights into the model's predictions. Clinicians can now understand the contribution of individual features, such as thyroxine levels, TSH, and

patient demographics, to the final diagnostic outcome. This level of transparency is invaluable in clinical settings, where decisions must be justified and aligned with established medical knowledge. For example, SHAP analysis revealed that "thyroxine", "thyroidsurgery", and "TSH" were among the most important predictors of thyroid conditions. These findings are consistent with clinical observations, as thyroxine and TSH are well-established biomarkers for thyroid function, while a history of thyroid surgery often indicates prior pathology. By quantifying the impact of these features, SHAP empowers clinicians to prioritize relevant clinical markers and tailor treatment strategies accordingly.

Another critical aspect of the experiments is the optimization of feature selection using the SSA. In clinical practice, datasets often contain a large number of features, many of which may be irrelevant or redundant. The SSA efficiently identifies the most informative features, reducing dimensionality and improving model interpretability without sacrificing performance. This approach ensures that the CAD framework focuses on clinically meaningful attributes, such as laboratory results and patient history, which are directly actionable in diagnosis and treatment planning. Additionally, the use of the TPE for hyperparameter tuning enhances the robustness of the models, ensuring that they perform optimally across diverse clinical scenarios.

Finally, the evaluation metrics used in the experiments (such as accuracy, precision, recall, specificity, and F1-score) are highly relevant to clinical decision-making. These metrics provide a comprehensive assessment of the model's performance, addressing both the correctness of predictions and the balance between sensitivity and specificity. The high accuracy scores achieved by the models, particularly those trained without data augmentation (*e.g.*, 95.06% for the DDTI dataset), underscore their potential to serve as reliable diagnostic aids in real-world clinical settings. Moreover, the rigorous cross-validation and confidence interval estimation ensure that the results are generalizable and robust, further enhancing their applicability in clinical practice. By bridging the gap between advanced AI techniques and clinical needs, this study paves the way for improved diagnostic outcomes and personalized treatment strategies in thyroid disease management.

## LIMITATIONS

While this study presents a comprehensive CAD framework for thyroid disease diagnosis, it has few limitations that should be acknowledged. First, the framework's performance was evaluated using only two datasets: the Thyroid Disease Patient dataset and the DDTI: Thyroid Ultrasound Images dataset. This limited scope may restrict the generalizability of the findings to broader populations and diverse clinical settings. Future work should incorporate additional datasets from varied demographic groups and healthcare systems to ensure robustness across different contexts.

Second, although the integration of numerical data and ultrasound images represents a significant advancement, the study did not incorporate other potentially valuable data modalities, such as genetic markers, clinical notes, or histopathological reports. These additional data sources could provide deeper insights into thyroid conditions and further

enhance diagnostic accuracy. For instance, genetic markers like BRAF mutations or RET/PTC rearrangements are known to play a critical role in thyroid cancer, and their inclusion could improve the framework's ability to identify malignancies.

Third, the study focused primarily on static data, neglecting temporal dynamics that could be crucial for understanding disease progression. Longitudinal data, such as repeated measurements of thyroid function tests over time, could offer valuable information about changes in patient status and help refine predictive models. Additionally, the lack of real-world clinical validation in this study highlights the need for prospective trials to assess the framework's effectiveness in practical healthcare environments.

Finally, while SHAP-based explainability enhances transparency, the interpretability of AI models remains an ongoing challenge. The current framework relies heavily on domain expertise to validate SHAP outputs, which may not always be feasible in clinical practice. Further advancements in XAI techniques, such as integrating causal inference or counterfactual analysis, could address these limitations and provide even more actionable insights for clinicians.

Addressing these limitations in future research will be essential to enhance the scalability, adaptability, and clinical applicability of the proposed CAD framework.

## CONCLUSIONS AND FUTURE DIRECTIONS

This study presented a comprehensive CAD framework for thyroid disease, integrating numerical patient data and ultrasound images to enhance diagnostic accuracy and interpretability. By utilizing advanced technologies such as ViTs and SHAP, the framework demonstrated significant improvements in thyroid disease diagnosis. The SSA was employed for optimized feature selection, while the TPE was utilized for hyperparameter tuning, resulting in superior performance compared to existing methodologies. The experimental results underscored the efficacy of the proposed framework, particularly in accurately classifying thyroid conditions. Notably, models trained without data augmentation consistently outperformed their augmented counterparts, achieving accuracy scores of up to 95.06% on ultrasound images and 99.71% on numerical data. These results highlight the robustness and reliability of the framework in handling diverse data modalities, making it a powerful tool for clinical decision support and treatment planning. The integration of XAI techniques, such as SHAP, provided transparent and interpretable diagnostic results, fostering trust and collaboration among healthcare practitioners and patients. By identifying key diagnostic features (*e.g.*, thyroxine, TSH, and thyroidsurgery), the framework not only improved diagnostic accuracy but also enhanced the understanding of thyroid disease pathology. This transparency is crucial for facilitating patient–centered care and improving health outcomes.

## FUTURE DIRECTIONS

Future research could explore advanced image analysis techniques like 3D CNNs and GANs to detect subtle thyroid abnormalities, integrate additional data modalities (*e.g.*, genetic markers, clinical notes) for personalized diagnosis, and extend the framework

to cross-disease analysis. Clinical validation across diverse populations, federated learning for data privacy, and enhanced SHAP-based explainability are essential for real-world adoption. Longitudinal studies will assess long-term diagnostic accuracy and clinical impact, ensuring the framework's scalability and effectiveness in healthcare.

### Funding
The authors received no funding for this work.

### Competing Interests
The authors declare that they have no competing interests.

### Author Contributions
- Saleh Ateeq Almutairi conceived and designed the experiments, performed the experiments, analyzed the data, performed the computation work, prepared figures and/or tables, authored or reviewed drafts of the article, and approved the final draft.

### Data Availability
The Thyroid Disease Patient Dataset is available at Kaggle: https://www.kaggle.com/datasets/kapoorprakhar/thyroid-disease-patient-dataset.

The DDTI: Thyroid Ultrasound Images Dataset is available at Kaggle: https://www.kaggle.com/datasets/dasmehdixtr/ddti-thyroid-ultrasound-images.

The code is available at GitHub and Zenodo:

- https://github.com/salehAteq/Integrating-AI-driven-CAD-Framework-with-Numerical-Data-and-Ultrasound-Images/tree/main

- salehAteq. (2025). salehAteq/Integrating-AI-driven-CAD-Framework-with-Numerical-Data-and-Ultrasound-Images: Paper Release (Manuscript). Zenodo. https://doi.org/10.5281/zenodo.15712614.

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
