# Peer review of "Advancing thyroid diagnosis: integrating AI-driven CAD framework with numerical data and ultrasound images"

_PeerJ Computer Science, doi:10.7717/peerj-cs.3063_

## Round 0.1 · original submission · Minor Revisions

Dear Authors,
Your paper has been revised. It needs minor revisions before being accepted for publication in PEERJ Computer Science. More precisely:

1) You should include test results on false positive and false negative rates for comparison with human expertise for subsequent personalized treatment decision-making. This would enhance the patient experience and increase trust in the use of AI techniques to reduce decision error rates.

2) You should provide more detailed information on how ViTs were trained (e.g., pretraining details, fine-tuning parameters, learning rate, number of epochs, regularization).

3) You should consider releasing code or pseudocode and specifying the computing infrastructure (e.g., GPU specs) to ensure reproducibility.

Reviewer 1 ·

Basic reporting

The basic reporting has a logical flow, and readers do not need in-depth thyroid domain-specific knowledge to follow.

Experimental design

The experimental design follows the standard approach: training, testing, and validating phases. The choice of a non-linear hyperparameter function was novel but needs enhancement on the use of SHAP to enhance the interpretability of the algorithms used.

Validity of the findings

No further comments as the validation results are impressive since DDTT open access data sets have been used and compared with no data augmentation.

Additional comments

The writing was clear, smooth, and could be easily followed. Perhaps test results on false positive and false negative rates should be included for comparison with human expertise for subsequent personalised treatment decision making. This would help for patient experience and trust in the use of AI techniques to reduce decision error rates.

Reviewer 2 ·

Basic reporting

The manuscript is generally well-written and professionally presented. The English language used is clear, formal, and unambiguous. The introduction clearly outlines the motivation for the study, emphasizing the need for integrating numerical data and imaging in thyroid disease diagnosis.

The background and literature review are appropriate and cite relevant prior work, although the manuscript could benefit from more references to recent Vision Transformer (ViT)-based medical imaging studies. The structure conforms to academic norms and is easy to follow.

One minor issue is the occasional formatting inconsistency in the abstract (e.g., a broken hyphen in “cuttingedge”), which should be corrected during revision.

No major concerns regarding basic reporting.

Experimental design

The study aligns with the journal’s scope and employs a sound methodological framework. The investigation is thorough and uses two relevant datasets (a structured patient dataset and the DDTI image dataset). The integration of ViTs, SHAP, SSA, and TPE creates a robust diagnostic pipeline.

However, several methodological clarifications are needed to improve reproducibility and transparency:

1、The manuscript should provide more detailed information on how ViTs were trained (e.g., pretraining details, fine-tuning parameters, learning rate, number of epochs, regularization).

2、 More discussion is needed on preprocessing steps for both the numerical and image data. The decision to exclude data augmentation (and its observed negative effect) warrants further explanation.

3、 The authors should consider releasing code or pseudocode and specifying the computing infrastructure (e.g., GPU specs) to ensure reproducibility.

Validity of the findings

The findings are plausible and supported by the reported results. Performance metrics (accuracy, precision, recall, F1 score) are appropriate, and the use of both 5-fold and 10-fold cross-validation adds rigor.

Nonetheless, there are areas that need attention:

No statistical significance testing is presented. Confidence intervals or statistical tests (e.g., t-tests or ANOVA) would help validate claims, particularly the claim that non-augmented data consistently outperform augmented data.

1、Since the system combines several components (SSA, TPE, SHAP, ViT), it would be beneficial to isolate and quantify the impact of each.

2、While SHAP is included, no visualizations are provided. Example SHAP plots or ViT attention maps would enhance understanding of model decisions and support the explainability claim.

The conclusions are mostly well-aligned with the results, though the paper would be stronger if limitations and possible clinical deployment challenges were discussed more explicitly.

Additional comments

This paper addresses an important healthcare problem and demonstrates how state-of-the-art AI techniques can be integrated into diagnostic frameworks. With minor but essential revisions to the methodology and presentation of results, the manuscript would meet PeerJ’s publication standards.

---

## Round 0.2 · accepted · Accept

Dear Author,
Your paper has been revised. It has been accepted for publication in PEERJ Computer Science. Thank you for your fine contribution.

Reviewer 2 ·

Basic reporting

The article is written in clear and professional English, and the overall structure conforms to PeerJ standards. The introduction successfully establishes the background and motivation for the work, emphasizing the challenges in thyroid disease diagnosis and the need for multimodal solutions.

The literature review is reasonably comprehensive, although a few recent studies comparing CNN-based and Transformer-based approaches for medical imaging could further contextualize the authors’ choice of ViTs. Definitions are generally adequate, although some technical terms (e.g., SSA, TPE) could benefit from brief intuitive explanations for interdisciplinary readers.

The results are reported clearly, but visual aids such as diagrams summarizing the proposed framework or comparative bar charts of key metrics would significantly improve clarity.

Experimental design

The study lies well within the journal's scope, and the experimental setup adheres to a high technical standard. The methods for data preprocessing, feature selection (SSA), hyperparameter tuning (TPE), and interpretability (SHAP) are all appropriate and modern.

Validity of the findings

The results are promising, with high performance metrics that support the authors’ claims. The paper rightly emphasizes interpretability using SHAP, which adds trust to the model’s predictions. Key features such as thyroxine and TSH are identified, which aligns well with clinical knowledge.

Additional comments

This paper presents a strong, interpretable, and multimodal CAD system for thyroid disease diagnosis, integrating cutting-edge techniques. Its contribution is meaningful, particularly in clinical AI, where explainability and multimodal fusion are critical.

While the results are compelling, a few key areas—such as baseline comparisons, architectural justification, and generalizability—should be addressed for the paper to be fully convincing. These can be handled during the rebuttal phase and do not require new data collection.